# Self-Challenging Language Model Agents

**Yifei Zhou**[*]
UC Berkeley

**Sergey Levine**
UC Berkeley

**Jason Weston**
FAIR, Meta

**Xian Li**[†]
FAIR, Meta

**Sainbayar Sukhbaatar**[†]
FAIR, Meta

## Abstract

Large language models are quickly becoming the foundation for intelligent agents that are capable of using tools. However, training such agents is challenging because it requires human creation and annotation of a diverse set of tasks, tools, and evaluation criteria. In this paper, we propose the *Self-Challenging* framework for training an agent on high-quality tasks that are generated by itself. The agent first plays the role of *challenger* and generates a task after interacting with the given tools. The tasks take the form of a novel general class of problems termed Code-as-Task, which are defined by an instruction, a verification function and solution and failure cases which serve as tests, allowing to filter only for high-quality tasks. The agent then takes an *executor* role and trains on those tasks with reinforcement learning using the evaluation feedback as a reward. Evaluation on two existing multi-turn tool-use agent benchmarks, M³ToolEval and TauBench, shows the *Self-Challenging* framework achieves over a two-fold improvement in Llama-3.1-8B-Instruct, despite using only self-generated training data.

## 1 Introduction

Large language models (LLMs) have demonstrated remarkable capabilities across a wide array of complex tasks [20, 14, 43], positioning them as promising agents for autonomous decision-making in open-ended environments, such as multi-turn tool use [46, 35] and GUI navigation [1, 41, 50]. Reinforcement learning (RL), which is already an integral part of post-training of LLMs, has emerged as a powerful tool for enhancing multi-turn agentic capabilities for LLMs by directly optimizing the final objective through interaction-based feedback [52, 53, 1, 24, 51].

An essential ingredient for applying RL to train a generally capable LLM agent is a large pool of high-quality tasks for agents to learn by trial and error [51, 22]. However, relying on human annotators for task creation is inherently costly, labor-intensive, and ultimately not scalable [51, 38, 47], thus underscoring the need for an automatic and reliable pipeline for synthesizing tasks. While prior works have built environment-specific pipelines for synthesizing tasks to perform RL, such as collaborative designs [53], web navigation [51], and fixing Github issues [22, 14], a *general task synthesis pipeline* for multi-turn LLM agents [37] in open-ended and tool-rich environments is still missing.

To come up with a synthetic task generation pipeline for general multi-turn tool-use LLM agents, we can start by reflecting on how human annotators design tasks given access to a set of tools and the environment. They will first try the tools and interact with the environment to see what kinds of goals are possible to achieve and what would happen if these goals were achieved. The tasks would then be sent to the LLM agent to see if it can achieve the goals set by the human annotators.

Motivated by this intuition, we propose a *Self-Challenging Agent* framework to construct synthetic tasks for training general multi-turn LLM agents at scale. In this framework, shown in Figure 1, the agent performs two distinct roles. In the *challenger* role, the agent first interacts in an unknown environment with tools to gather information and generate possible tasks. Subsequently, these

---

[*]Work done at FAIR, Meta. Correspondance to yifei_zhou@berkeley.edu.
[†]Equal advising.

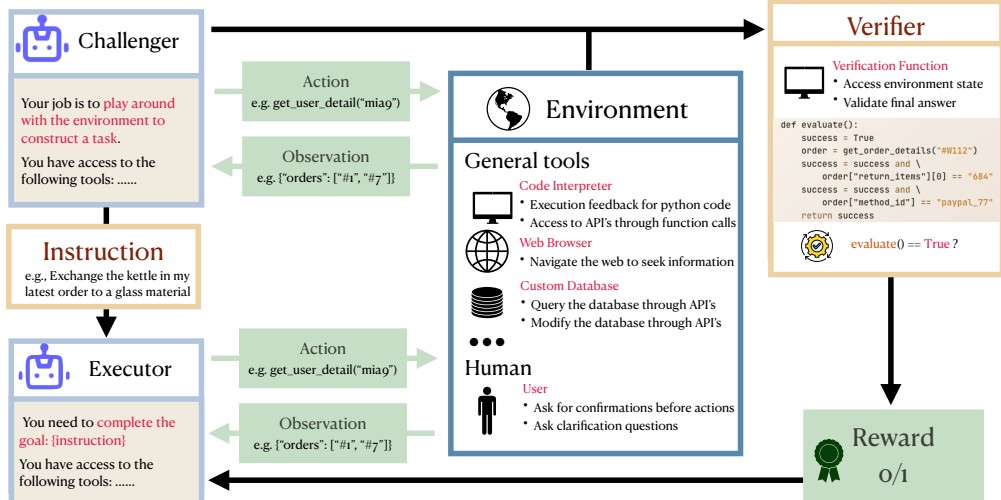

Figure 1: **Overview of Self-Challenging Agent**. The agent takes on two roles: task challenger, and task executor. The task challenger proposes a task along with a verification method to verify the solution to the task. The task executor generates a solution and obtains a reward from the environment based on the verification method.

synthetic tasks are tackled by the agent in the *executor* role, who attempts to accomplish them by interacting with the same environment. However, ensuring generated tasks are (1) **feasible**, (2) **verifiable**, and (3) **difficult** becomes increasingly complex in such scenarios. If the challenger agent inadvertently generates flawed or impractical tasks, it risks introducing substantial noise into the training data, thereby contaminating and destabilizing the learning process of the executing agent.

To enable the creation of high quality generated tasks we introduce the *"Code-as-Task"* (CaT) formalism, wherein each generated task comprises four explicitly defined components: an instruction, a verification function, an example solution, and explicitly enumerated failure cases, where all components except the instruction are expressed in code due to its generality and expressivity [44, 15, 37]. This structured representation ensures task feasibility, as exemplified by provided solutions, task difficulty, as exemplified by typical failure cases, and task verifiability through rigorous, executable verification criteria. By leveraging external code executors, our framework automatically filters out erroneous or impractical tasks, significantly mitigating risks associated with low-quality tasks.

Our experiments are conducted in four different tool-use environments from M$^3$ToolEval [37] and Tau-Bench [46] spanning tool-based calculations, web browsing, retail services, and flight booking. We apply our method to generate synthetic tasks and rely purely on these synthetic tasks to fine-tune the LLM agent before evaluating on the existing out-of-distribution test tasks. Empirically, we establish the advantages of our Self-Challenging Agent (SCA) framework in two important settings: distillation, where the goal is to distill the expertise of a stronger model to a weaker model without any existing tasks, and self-improvement, where in the absence of a stronger model the weaker model needs to supervise itself to make progress. For distillation, we are able to improve the student model Llama-3.1-8B-Instruct [34] by 20.2% in terms of absolute average success rate across all 4 environments without any existing tasks. Even in the absence of a stronger model, through RL training on its own synthetic tasks and rollout trajectories, SCA training is able to double the success rate from 12.0% to 23.5%, outperforming the prior state-of-the-art self-improvement method for LLM agents by a wide margin.

## 2 Related Work

**Self-Challenging agents.** The idea of an agent self-synthesizing tasks has been explored before the emergence of LLMs. Notably, Asymmetric Self-Play (ASP) [33, 21] lets one agent act in the environment first, followed by the second agent who aims to reach the same end state. The adversarial nature of their reward encourages the first agent to act in ways that are difficult to imitate. While

this self-play idea has been historically successful in relatively constrained domains like robotics simulations [3, 33, 21], math reasoning [6], or even to single-turn instruction following tasks [4, 47], it has not been studied in terms of how to generalize ASP to open-ended and partially observable tool-use agent environments while preserving task quality. A concurrent work [49] adopted ASP to generate coding tasks with execution verification. In this work, we propose the Code-as-Task formalism and thus utilize external code executors to ensure the synthetic tool-using tasks are feasible, verifiable, and difficult.

**Autonomous improvement of LLM agents.** Recent studies have found that LLMs can be prompted to evaluate the outcomes of LLM agents by examining their trajectories [23, 9]. Using autonomous LLM evaluations as reward feedback, RL [1, 36, 2] or hindsight relabeling [19, 18, 5] can be applied to improve the performance of LLM agents without human supervision. This self-improvement loop can be further enhanced through self-generated instructions [38] to enhance the diversity of the skills acquired [51, 32, 10, 24, 48, 7, 11, 5]. However, while autonomous evaluations and task proposers can work well for relatively basic tasks, they do not have explicit mechanisms to improve task quality and ensure its verifiability. In contrast, our work introduces a novel framework for a task challenger agent to explore the environment before generating tasks and grounds the tasks in the Code-as-Task format to significantly improve the task quality by leveraging an external code executor. In our experiments, we show that these design decisions lead to significant improvements in both the distillation and self-improvement settings.

**Code-as-action for tool-use LLM agents.** Equipping LLMs with external tools can not only augment text-generation and information-gathering capabilities [27, 25, 17, 8, 42, 30] but also enable them to take actions in the real world [46, 35, 15, 37]. While early explorations have designed heterogeneous interfaces for LLM agents to use tools such as JSON or text [16, 45, 25], recent studies have found that code can be used as a unified general interface for LLM agents to interact with API functions [35, 44, 37], navigate through the web [31], make travel plans [40], and even control robots in the physical world [15, 12, 13]. However, using code as a universal interface for LLMs to autonomously generate synthetic tasks has not been studied. Our work proposes a novel Code-as-Task format that formalizes tasks consisting of instructions, verification functions, example solutions, and failure cases, to ensure the quality of the tasks in open-ended, tool-rich LLM agent environments.

## 3 Problem Setup

In the standard LLM agent setting, the agent has the goal of *executing* a given task. In this executor role, the agent interacts with the environment that contains multiple tools, such as a code interpreter, web browser, and even human users, as shown in Figure 1. Let us represent the agent by the executor policy $\pi^{\text{exec}}$. At step $t$, the policy will output an action $a_t$, which can be a piece of code calling a tool API, or a question to the user. Then the response from the tool or the user $o_{t+1}$ will be observed and appended to the context of the LLM when sampling the next action

$$a_{t+1} \sim \pi^{\text{exec}}(\cdot|o_{0:t+1}, a_{1:t}), \quad s_{t+1} \leftarrow \mathcal{T}(s_t, a_t). \tag{1}$$

The action will also have an effect on the environment and change its state $s_t$ (i.e. database entries), represented by a transition function $\mathcal{T}$ here. Here the initial observation $o_0$ can contain useful information like API documentation. It can also contain instructions about the task $c \in \mathcal{C}$, which can be further clarified by conversing with the user. Once the agent completes the task after $T$ steps, a verification method, e.g. a function that checks the state of the environment and the solution from the agent, will be executed to evaluate the success of the solution and issue a reward to the agent: $R_c(s_T, a_T) \in \mathbb{R}$. As in principle the interactions between a language model agent and any external environment can be considered as a form of tool use, this formulation of multi-turn tool-use language agents encompasses most LLM agent tasks [37], such as agentic coding [14, 22], web browsing [31, 50], and even interactions with users [53, 46]. In this paper, however, we assume an unsupervised training setup where we do not have access to the ground-truth task set $\mathcal{C}$ and its verification functions $R$. Instead, Self-Challenging allows the LLM agent to construct its own tasks and their evaluation functions for RL training. We will detail our method in the following section.

## 4 Self-Challenging Agents

We now describe our Self-Challenging framework for training LLM agents. Different from the standard LLM agent setting, our method enables agents with two roles: (1) As a *task challenger*, it

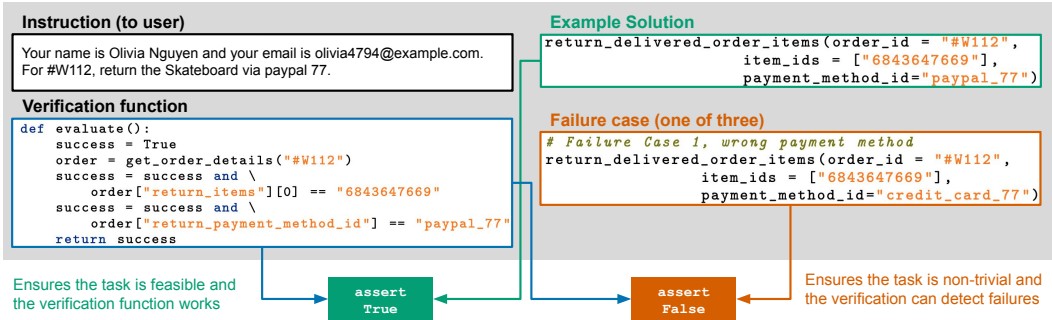

Figure 2: An example of a synthetic **Code-as-Task (CaT)** generated by the task challenger, in a TauBench-based environment [46]. The task challenger interacts with the environment taking a series of actions calling different tools to gather information, before generating the synthetic task, consisting of an instruction, verification function, example solution and failure cases (shortened in the figure for brevity). Automatic filtering is applied to CaTs to keep only valid tasks where the example solution can pass the verification function and the failure cases cannot.

interacts with the environment and self-synthesizes tasks in our novel Code-as-Task format in order to make high quality and challenging for training. (2) As a *task executor*, it is trained on these self-synthesized tasks in order to learn to solve real tasks at test time, for which there is less available annotated data. We will describe each in turn.

## 4.1 Task Challenger

In the task challenger role, the agent interacts with the environment with the end goal of constructing a viable task. Accessing the tools in the environment allows the challenger to gather information about available tools and probe what type of tasks can be performed. Let $\pi^{\text{task}}$ be the policy of the task challenger. Similar to $\pi^{\text{exec}}$, the challenger takes a series of actions calling different tools: $a_{t+1} \sim \pi^{\text{task}}(\cdot|o_{0:t+1}, a_{1:t})$. At the end, however, it must output a task specification that includes a verification function: $\hat{c}, \hat{R}_{\hat{c}} \sim \pi^{\text{task}}(\cdot|o_{0:T}, a_{0:T-1})$. In this manner, we collect a set of synthetic tasks $\hat{\mathcal{C}} = \{\hat{c}_i\}_{i=1}^N$, on which we can train the executor agent. While the distribution of the generated tasks $\hat{\mathcal{C}}$ might not exactly match the real-world tasks $\mathcal{C}$, training on $\hat{\mathcal{C}}$ could lead to improved performance on $\mathcal{C}$ as long as there are common similarities and shared skill-sets. In the next section, we will explain how the class of tasks are structured, and how evaluation is done to obtain rewards.

## 4.2 Code-as-Task (CaT) Class of Tasks

As described in the previous section, in order for the task challenger policy $\pi^{\text{task}}$ to generate high-quality tasks, $\pi^{\text{task}}$ itself also needs to be a reliable agent to take a series of actions to gather more information about the environment and the unobserved states. As this is not trivial especially if the tasks need to be challenging, we expect $\pi^{\text{task}}$ to sometimes generate tasks that are either infeasible or ambiguous. Incorporating these tasks for training the executor policy $\pi^{\text{exec}}$ will reduce the signal-to-noise ratio and contaminate the training process.

We thus propose the "Code-as-Task" (CaT) class of tasks which can help in filtering out such low quality tasks. As depicted in Figure 2, CaTs leverage code as an interface to automatically evaluate the quality of synthetic tasks. The basic components of a CaT synthetic task include an instruction and a verification function. The verification function $\hat{R}$ will provide a sparse 0/1 outcome reward based on whether the final state $s_T$ and answer $a_T$ passes the verification function. However, false negatives (i.e. the agent gets a reward of 0 without doing anything wrong) and false positives (i.e. the agent gets a reward of 1 without actually fulfilling the request) can easily occur when $\pi^{\text{task}}$ hallucinates during task proposals. To address these issues, we introduce additional components: an example solution and three failure cases. These are automatically checked against the verification function to ensure that tasks are feasible yet non-trivial and the verification function works properly. We observe in experiments that this makes our proposed tasks significantly higher quality. We include a specific prompt (see Appendix Figure 9 for an example) as an input to $\pi^{\text{task}}$ so that it will adhere to CaT and output all four components. A full output example of CaT is provided in Appendix Figure 15.

### 4.3 Task Executor

Once high quality tasks for the open-ended environment are generated via the CaT formalism, they can then be directly used for training the task executor. We explore two possibilities in how these tasks can be helpful for training the executor agent. We first discuss the self-improvement setting where $\pi^{\text{exec}}$ improves itself with its own rollout trajectories through RL. We also explain an additional possibility of distillation in the presence of a stronger teacher model via the synthetic tasks.

**Self-improvement.** We consider the self-improvement setting where both $\pi^{\text{exec}}$ and $\pi^{\text{task}}$ are the same LLM. In this setting, after deriving a set of high-quality tasks $\hat{\mathcal{C}}$ through CaT generation, we use the same LLM $\pi^{\text{exec}}$ to collect trajectories and annotate them with rewards calculated from verification functions $\hat{R}$ included in the tasks and apply RL for optimizing the policy. In particular, we applied one-step REINFORCE [39], the most basic RL optimization method, that minimizes:

$$\mathcal{L} = -\sum_{\hat{c} \in \hat{\mathcal{C}}} \mathbb{E}_{a_{1:T} \sim \pi^{\text{exec}}} \left[ \hat{R}_{\hat{c}}(s_T, a_T) \sum_{t=0}^{T} \log \pi^{\text{exec}}(a_t | o_{0:t}, a_{0:t-1}) \right]. \tag{2}$$

Because of our reward structure with 0/1 outcome reward, this RL optimization objective is essentially equivalent to performing supervised finetuning (SFT) on successful trajectories (i.e. $\hat{R}_{\hat{c}}(s_T, a_T) = 1$) only, i.e. Rejection Fine-Tuning.

**Distillation.** We then consider an additional possible use of the synthetic tasks to distill the domain-specific knowledge of a stronger LLM into a weaker LLM in a specific open-ended environment without any additional human data. This setting could be particularly useful when we have a larger generalist LLM that would be too costly or too slow to deploy, so we would like an automatic process to distill the capability of the larger model in this environment to a smaller LLM that would be cheaper and faster to deploy. In this setting, after creating a set of high-quality tasks $\hat{\mathcal{C}}$ through CaT generation, we use the stronger LLM to sample a dataset $\hat{\mathcal{D}}$ containing trajectories $\{o_{0:T}, a_{0:T}\}$ on those tasks. We then apply SFT with the cross entropy loss for training the weaker student LLM:

$$\mathcal{L} = -\sum_{\{o_{0:T}, a_{0:T}\} \in \hat{\mathcal{D}}} \left[ \sum_{t=0}^{T} \log \pi^{\text{exec}}(a_t | o_{0:t}, a_{0:t-1}) \right]. \tag{3}$$

While one alternative is to train only on successful trajectories, our preliminary experiment results show that even failed trajectories from a stronger model can be beneficial for a weaker model to learn from, such as reflections on mistakes.

## 5 Experiments

We evaluate the *Self-Challenging Agent* (SCA) framework in actual tool-rich and open-ended LLM agent environments. Specifically, we use it to train an agent in four different environments and in both self-improvement and distillation settings using synthesized data only. The performance is then measured on actual real test tasks, and compared against baselines. Finally, we provide ablation and analysis experiments including data scaling results.

### 5.1 Environments

Our experiments are conducted in two multi-turn tool-use LLM agent benchmarks, featuring tasks from four different environments that come equipped with functional verifiers for reliable evaluations. In both environments, we instruct the agent to first output its thoughts and then follow them with a block of executable Python function calls or a response to a simulated human user within the environment. The execution results of the executable code or the simulated response from the user serve as the observations for the next step. We limit the maximum number of steps in each environment to be 15. The prompts used to apply our method to these environments are included in Appendix D and more details about the environments can be found in Appendix A.

**M³ToolEval [37]** is a multi-turn function-calling benchmark where the success of each task is determined by pattern-matching the agent's final answer with the reference solution. We report the performance in both the Calculation and Web Browsing environments with existing test tasks from M³ToolEval. The Calculation environment involves using tools from a Travel Planner, DNA Sequencer, Message Decoder, and Trade Calculator (approximately six tools in each domain) to perform calculations. To use SCA in this environment, the task challenger applies the given tools to create new tasks, generates the verification function to be pattern-matching with a reference solution,

Table 1: **Main results.** We validate the effectiveness of Self-Challenging Agents (SCA) in both the distillation and self-improvement settings compared to zero-shot LLMs and PAE [51] baselines. For both settings, we generate 800 synthetic tasks and 12k offline rollout trajectories. Pass@1 results are averaged over four independent trials, and pass@4 is calculated from the same four trials. The best results in each setting are in bold.

| | M$^3$ToolEval | | | | Tau-Bench | | | | Average | |
| | Calculation | | Web Browsing | | Retail | | Airline | | | |
| | Pass@1 | Pass@4 | Pass@1 | Pass@4 | Pass@1 | Pass@4 | Pass@1 | Pass@4 | Pass@1 | Pass@4 |
|---|---|---|---|---|---|---|---|---|---|---|
| *Zero-Shot* | | | | | | | | | | |
| GPT-4o | **72.4** | **85.4** | 52.9 | 70.6 | 18.7 | 37.4 | **15.8** | **36.7** | 40.0 | 57.5 |
| Llama-3.1-70B | 57.3 | 79.2 | **64.0** | **79.4** | **34.6** | **56.5** | 9.2 | 20.0 | **41.3** | **58.8** |
| Llama-3.1-8B | 20.3 | 43.8 | 16.2 | 42.1 | 8.9 | 15.7 | 2.5 | 10.0 | 12.0 | 27.9 |
| *Distillation* | | | | | | | | | | |
| Llama-3.1-8B-PAE | **44.3** | **77.1** | 45.6 | 76.5 | 23.7 | 40.9 | **6.7** | 13.3 | 30.1 | 52.0 |
| Llama-3.1-8B-SCA | 43.2 | 72.9 | **50.0** | **82.4** | **28.9** | **48.7** | **6.7** | **23.3** | **32.2** | **56.8** |
| *Self-Improve* | | | | | | | | | | |
| Llama-3.1-8B-PAE | 27.6 | 54.2 | 14.0 | 38.2 | 9.8 | 18.3 | 0.0 | 0.0 | 12.9 | 27.7 |
| Llama-3.1-8B-SCA | **31.8** | **62.5** | **44.9** | **67.6** | **13.0** | **25.2** | **4.2** | **10.0** | **23.5** | **41.3** |

provides an example solution as a series of operations that can arrive at the reference solution, and defines failure cases as incorrect answers.

**Tau-Bench [46]** is a multi-turn customer service environment where the LLM agent needs to interact with a user (simulated by GPT-4o [20]), query the database, and make corresponding modifications to fulfill the user requests. It is composed of Retail and Airline environments corresponding to customer support for E-commerce service and flight booking services. Around 15 tools such as "book_flight" and "exchange_order_items" are available for each environment. When a user response is needed, we use the same model as the backbone of the agent (Llama-3.1-8B-Instruct) to simulate the user instead of GPT-4o as used at final evaluation time.

### 5.2   Main Comparisons

**Experiment Setup.** We conduct our experiments by RL fine-tuning Llama-3.1-8B-Instruct ("Llama-3.1-8B" for short) [34] as the base model $\pi^{\text{exec}}$. In both distillation and self-improvement settings, we use the same fixed Llama-3.1-8B model as the task challenger $\pi^{\text{task}}$. In the self-improvement setting, all training tasks and trajectories are generated by Llama-3.1-8B itself. In the distillation setting, we use larger Llama-3.1-70B-Instruct ("Llama-3.1-70B" for short) as the stronger model to sample demonstration trajectories. We compare SCA with several baselines in different settings. Firstly, we include the zero-shot performance of **Llama-3.1-8B** and **Llama-3.1-70B** to study how much performance gain SCA can produce from the pre-trained checkpoint, and the zero-shot performance of **GPT-4o** as a reference for advanced proprietary models. We also compare to the Proposer-Agent-Evaluator (**PAE**) [51] baseline, a prior state-of-the-art for task synthesis and autonomous improvement, which has been shown to be effective for open-ended GUI navigation agents. In PAE, instructions are proposed autonomously by prompting Llama-3.1-8B with the initial observation, such as the API documentation of available tools or the initial web page for web browsing. Success is also judged by prompting the same model, Llama-3.1-8B, with the instruction and the rollout trajectory. The prompts for both the PAE task proposer and evaluator are included in Appendix D. The crucial differences between PAE and **SCA** are: (1) the task proposer in PAE generates tasks directly from API documentation and the initial observation instead of serving as an agent that actively interacts with the environment to gather information before creating the task; (2) PAE only generates instructions while our CaT tasks contain instructions, verification functions, example solutions, and failure cases; and (3) PAE prompts the same model to serve as the judge instead of relying on verification functions as in SCA. Note that the third point only makes a difference in the self-improvement setting because all demonstration trajectories, regardless of success or failure, are used for the distillation setting.

**Distillation Results.** The main results in the distillation setting are presented in Table 1. We first note the surprising generalization to out-of-distribution test tasks entirely through training on synthetic tasks. Both PAE and SCA are able to achieve more than 18% and 20% absolute improvement in the average success rate, respectively. This is especially noteworthy given the striking difference between synthetic tasks and real tasks. However, we also observe major limitations of PAE. For example,

PAE can only propose very ambiguous tasks based on the limited initial information it gathers from the environment. While PAE works with distillation, particularly in more straightforward and fully observable environments such as Calculation with an average improvement of +23% Pass@1 success rate, its tasks are mostly limited to the information from the initial observation and do not incentivize the teacher model to provide demonstrations with a wide coverage over the environments when it is partially observable. In these environments, we find the task challenger agent in SCA to be superior where it can actively explore the environment to generate more precise tasks that incentivize the teacher model to sufficiently explore the environment. We find that this leads to better performance compared to PAE in all 3 partially observable environments, with the performance gain as large as 5.2% in terms of Pass@1 success rate in Retail. As we will see in the next section, these weaknesses in PAE's proposed tasks are a larger problem in the self-improvement setting, when a stronger teacher model is not available.

**Self-improvement Results.** The main results for the self-improvement setting are also presented in Table 1. We find that the quality of synthetic tasks matters much more in this setting compared to the distillation setting. While PAE is still effective in the fully observable Calculation environment, achieving +7.3% improvement in Pass@1 success rate, it can only achieve marginal improvements or even decreased performance compared to the baseline in the other three partially observable environments. In particular, the autonomously proposed tasks from PAE in Airline actually result in a -10% drop to 0% Pass@4. In contrast, thanks to the diversity and preciseness of the tasks generated by the task challenger agent and the accurate reward feedback from the Code-as-Task formulation, SCA is able to achieve better performance both in the fully observable Calculation environment and the other three partially observable environments, with an average of +11.5% gain compared to the base model and a +10.6% advantage compared to PAE in terms of Pass@1. This shows the advantage of SCA in general tool-rich and open-ended environments, over the prior state-of-the-art PAE, whose utility crucially depends on the fully observable assumption of the environment.

### 5.3 Analysis and Ablations

**How do different RL algorithms work with SCA?**
While the main self-improvement results in Table 1 are achieved through simple Rejection Fine-Tuning on successful offline rollout trajectories as described in Section 4.3, in principle the design of SCA should be compatible with any RL algorithm, even the state-of-the-art online RL algorithms for LLM such as PPO [28] and GRPO [29]. We investigate the effectiveness of SCA when training with different RL algorithms and report the results in Figure 3, including offline methods such as Rejection Fine-Tuning and DPO [26] and online methods such as GRPO and PPO. While offline methods such as Rejection Fine-Tuning and DPO can already achieve significant self-improvements from 20.3% to 31.8% Pass@1 success rate, online RL on the same tasks can push the performance to another level of 43.2% Pass@1. However, the additional performance gain comes at the cost of increased infrastructure complexity to accommodate efficient on-policy sampling and instability to hyperparameters, as we observe that the performance of GRPO suddenly drops to 0% at the end of training when it is not tuned carefully.

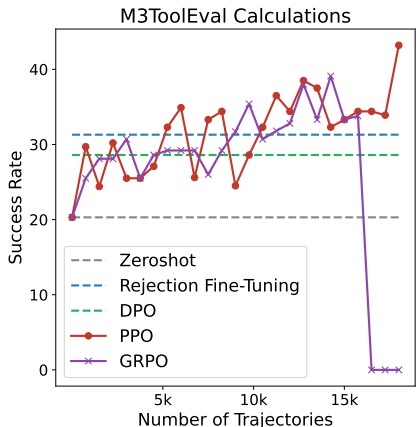

Figure 3: **Ablation studies of different RL algorithms** with synthetic tasks generated from SCA, in the Calculation environment from M$^3$ToolEval. Pass@1 success rates are reported. We find that online RL algorithms in general attain even better performance on out-of-distribution test sets, but they are more unstable and require more careful tuning.

**What is the impact of the different components in CaT?** We perform human annotation experiments to understand the importance of different components in CaT. Specifically, we compare four different variants for generating tasks, including the PAE baseline, CaT w/ Instruction + Verification Function Only, CaT w/ Instruction + Verification Function + Solution, and CaT (Instruction + Verification Function + Solution + Failure Cases), all using Llama-3.1-8B. For CaT w/ Instruction + Verification Function Only, we perform an automatic filtering to only keep tasks where the verification function is runnable Python code. After filtering, 47.7% of tasks remain with this variant. For CaT w/ Instruction + Verification Function + Solution, we perform an automatic filtering to keep only tasks

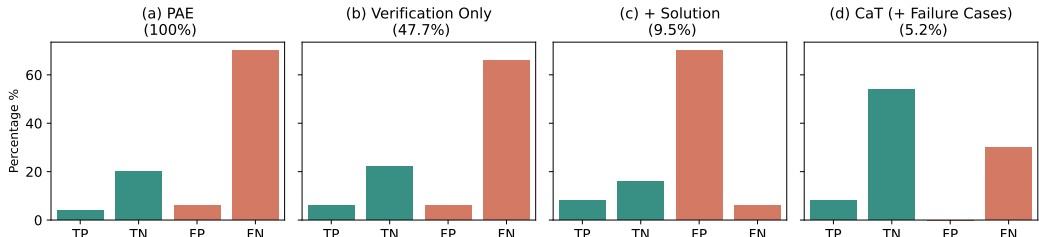

Figure 4: **Human annotations of synthetic task qualities.** 50 rollout trajectories from Llama-3.1-8B in the Retail environment from attempted synthetic tasks from each variant are manually labeled to fall into one of the four categories including False Negative (FN), False Positive (FP), True Negative (TN), and True Positive (TP). The pass rates of the task challenger generating a task passing all filters for each category are shown in parentheses. We observe that CaT can significantly reduce both FN and FP, which are invalid tasks or wrongly labeled trajectories.

where the solution can pass the verification function. The percentage of passing tasks for this variant is 9.5%. For CaT (Instruction + Verification Function + Solution + Failure Cases), we perform all automatic filtering steps to keep only tasks where the solution can pass the verification functions while none of the failure cases can pass. The percentage of passing tasks for full CaT is 5.2%.

For each variant, we obtain 50 passing synthetic tasks and collect 50 rollout trajectories from Llama-3.1-8B in the Retail environment. For each rollout trajectory, one of the authors manually inspects the trajectory and the task to classify it into one of the four categories: **(1) False Negative (FN)**, where the agent gets a reward of 0 because the task is impossible to be complete. For example, the agent fails the task of "return order #W000001" but it is impossible because this order does not exist. **(2) False Positive (FP)**, where the agent gets a reward of 1 but does not actually fulfill the instruction. For example, the agent might return an order instead of exchanging the order. **(3) True Negative (TN)**, where the agent gets a reward of 0 because it indeed makes a mistake in attempting the task. **(4) True Positive (TP)**, where the agent gets a reward of 1 and it indeed fulfills the instruction. The statistics from human annotations are presented in Figure 4.

Firstly, we notice that tasks from PAE result in a significant fraction of FN, with the main failure mode of generating ambiguous and impossible tasks for the agent to complete. Explicitly asking the task challenger to interact with the environment to gather more information and generate a verification function along with the instruction in Verification Only amounts to marginal improvements. We find that asking the task challenger to generate an example solution and enforce the requirement of the example solution passing the verification function can significantly reduce FN by eliminating impossible tasks, but it increases FP. The main failure mode is that the verification function is too lenient and sometimes even no actions can pass the verification function. While these overly lenient verification functions do not account for a large percentage initially, the filtering mechanism based on the example solution passing verification functions tends to amplify this distribution. Fortunately, we can apply a simple fix by adding another filtering mechanism based on failure cases to completely remove FP in CaT. However, there is still a large percentage of FN even in CaT. The main failure mode of FN in CaT is incomplete instructions where the instruction does not contain all necessary information to complete the task. For example, the instruction may be "Help me return one of my latest orders" without specifying which order needs to be returned. As this group of FN involves nuanced semantic dependencies, we leave it as an open question for future research.

**How does the automatic filtering mechanism in CaT affect task diversity and difficulty?** One possible concern is that the heavy automatic filtering mechanism might collapse the distribution of tasks and make them less diverse. To investigate this concern, we analyze the distribution of task difficulty before and after the filtering step of CaT in the Retail environment, where we measure task difficulty using the length of the example solution. As presented in Figure 5 (a), we have found that for Llama-3.1-8B, the distribution of task difficulty before the filtering step is more spread out, while after the filtering step it tends to be more homogeneous. However, the distribution after filtering remains similar with the stronger model Llama-3.1-70B shown in Figure 5 (b), where CaT merely refines the task distribution instead of collapsing it. This suggests that CaT filtering only removes poor generations typically found in weaker models while preserving the diversity of the distribution of valid tasks.

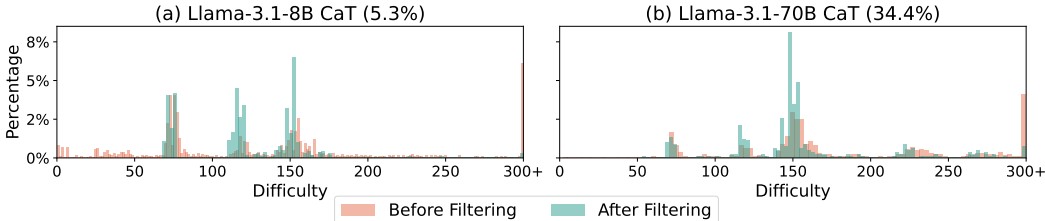

Figure 5: **Analysis of the distribution of task difficulty** before and after the filtering step of CaT in the Retail environment. The task difficulty is represented by the length of the example solution. The percentages of passing tasks after filtering are included in parentheses. We observe that CaT filtering can result in a less diverse task distribution for the less capable Llama-3.1-8B but preserves the original task distribution for the stronger Llama-3.1-70B model.

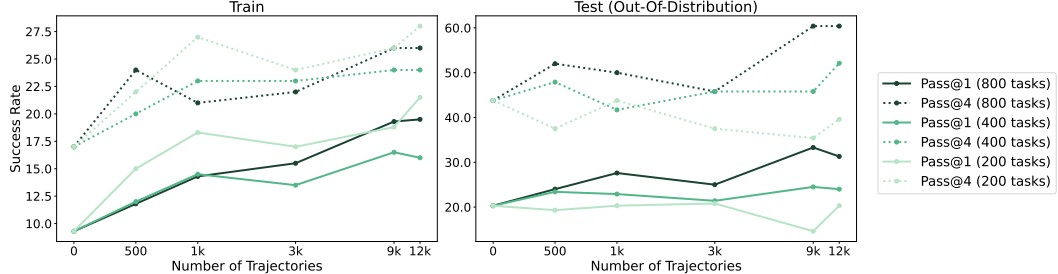

Figure 6: **Scaling analysis** for SCA with different number of tasks in M$^3$ToolEval Calculation. The results show that scaling the number of synthetic tasks is more effective compared to scaling the number of trajectories per task on the test data. In contrast, on the train data a smaller number of tasks but more trajectories gives a higher success rate, but this does not generalize out-of-distribution.

**Should we scale the number of tasks or the number of trajectories per task?** In order to further improve performance with more synthetic data, there are two axes along which we can generate more rollout trajectories: 1) generating more synthetic tasks to collect rollout trajectories, or 2) collecting more rollout trajectories for each synthetic task. Figure 6 presents our findings in investigating these two scaling axes in M$^3$ToolEval Calculation. Firstly, we notice that scaling the number of interactions always results in improvements in Pass@1 and Pass@4 success rates in the train set for all three different train set sizes. However, such improvements in the train set do not necessarily translate to improvements in the out-of-distribution test set. In particular, training on 200 tasks even slightly degrades the test set Pass@1 and Pass@4 performance, and training on 400 tasks only leads to marginal improvements. Only when the training set has sufficient diversity and coverage, as in the case of 800 tasks, can we see a steady improvement in terms of test performance as we collect more rollout trajectories. This result underscores the requirement of a large set of training tasks for general LLM agents and the promise of SCA to automate the process of large-scale task creation.

## 6 Broader Impact

The objective of this project is to empower LLM agents with the ability to autonomously enhance their capabilities through interactions with the environment, thereby reducing reliance on costly human supervision. Although the models trained in this paper exhibit inadequate performance and struggle with relatively simple tasks, such as booking flight tickets and processing returns, future self-improvement methods leveraging more advanced pre-trained LLMs may potentially yield models with superhuman capabilities. To mitigate potential risks, it is essential to conduct thorough research on aligning strong models with human values, ensuring that these values are preserved even as the models improve themselves to achieve superhuman performance levels.

## 7 Limitations

While our work shows the promise of an LLM agent generating high-quality tasks to improve itself, there are still limitations, which are also future research opportunities. First, despite the CaT formalism, there is still a non-trivial percentage of False Negative examples as shown in Figure 4.

The main failure mode is semantic nuances such as ambiguity or missing information that can be difficult to distinguish. This results in the suboptimality gap between RL training with SCA tasks and oracle tasks as shown in Appendix B. Additionally as detailed in Appendix B, because of the distinctions between different environments studied in this paper, the LLM agent tends to mostly improve its environment-specific skills instead of overall agentic capabilities. It remains an open research problem on how to enhance the environment-general agentic capabilities of LLM agents.

## 8 Conclusion

In this paper, we present the Self-Challenging Agent (SCA) method for self-improvement of general multi-turn tool-use LLM agents. SCA can create its own tasks to challenge itself and learn from them. To do this, it utilizes the Code-as-Task (CaT) formulation which ensures high quality synthetic tasks. Through RL on these self-generated synthetic tasks, SCA can be used to train a Llama-3.1-8B model to achieve an average relative success rate improvement of 95.8% on existing test tasks across four different multi-turn tool-use environments. Our results show the promise of self-improvement for general multi-turn tool-use LLM agents, without reliance on the manual process of creating a diverse set of tasks, tools, and evaluation criteria. While SCA serves as a preliminary step, there remains many research questions for building an effective self-improvement flywheel for general LLM agents.

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

# A Additional Environment Details

Due to space constraints of the main text, we provide additional details of the environment in this sections.

**M³ToolEval:** an example task in this environment is: "Decode a message that went through three steps: first, a Caesar cipher with a shift of 3; then reversed; and finally, encoded to hexadecimal. The final hex-encoded message is '726f77746e6153794d'." Compared to other environments, Calculation is fully observable, and almost all information about this environment is encoded in the initial observation containing the API documentation of each tool. The Web Browsing environment involves text-based navigation through a set of synthetic web pages to find specific information. Initial observable information in only contains the landing web page and the tools that the agent can use to navigate the website; more information must be gathered through explicit interaction with the environment.

**Tau-Bench:** an example task in this environment is "You are Fatima Johnson in 78712. You want to cancel all pending orders (since they are no longer needed) and return the watch you have received (but nothing else), and you want to know the total amount you can get back. You are a private person that does not want to reveal much about yourself,". In that case, the ground-truth verifier checks if the desired changes have been actually made in the database. When we apply our method to generate synthetic tasks for the tools and setting of this benchmark, we set the verification function to a python function that performs database queries to verify desired effects, the example solution to be a series of database modification calls, and the failure cases to be some close attempts. The initial observations for both environments only contain the API documentations for each tool, and any information regarding the database is only available to the agent through explicit function calling such as "get_reservation_details". We noticed 40% of the tasks in Airline are considered successful when no actions are taken so we removed them from our experiments to reduce noise.

**Environment modifications for task challenger:** each time when the task challenger tries to synthesize a task in CaT format, we will reset the hidden states of the environment (e.g. the flight and retail database) and run an automatic checker. The automatic checker verifies 1) all code is runnable, 2) the example solution can pass the verification function, and 3) the failure cases cannot pass the verification. If the automatic checker fails, it will return its error traceback to the task challenger to revise the task.

# B Additional Discussions

In this section, we present additional experiment results and analysis to understand the limitations of Self-Challenging Agent (SCA) and possibility for future research.

**How does the improvements from SCA transfer between multiple environments?** To understand whether the improvements from SCA can generalize across different environments, we compare the performance of training separate models for each individual environment and that of training a joint model with aggregate data from all environments. As shown in Table 2, we found marginal improvements, or even worse performances, of training a joint model on the data from all environments. By inspecting the trajectories from pretrained Llama-3.1-8B-Instruct and Llama-3.1-8B-SCA, we found that the main challenges faced by the agent in those environments are environment-specific e.g. a "book_hotel" tool in Calculation does not take position arguments or the agent should first ask for the user emails and zip code before checking the account details, and Self-Challenging Agent mainly improved the LLM agents' capabilities to get around these environment-specific challenges. It remains an open research problem if we can design a self-improvement method that enhances the environment-general agentic capabilities.

**How does the task quality from SCA compare with oracle tasks?** We conduct additional experiments to investigate how tasks generated by SCA compare with high-quality oracle tasks in M³ToolEval Calculation. To construct oracle tasks, we manually create 5 seed tasks for each domain in Calculation that resemble existing test tasks and prompt Llama-3.1-8B to generate similar tasks with detailed documentation describing the implementation of each tool. We then prompt

Table 2: **Cross-environment transfer analysis** where we compare the performance of training a separate model for each environment with that of training the same model with aggregate data from all environments. We found marginal, if not worse, benefits from aggregate training for all 4 environments.

| | M³ToolEval | | | | Tau-Bench | | | | Average | |
| | Calculation | | Web Browsing | | Retail | | Airline | | | |
| | Pass@1 | Pass@4 | Pass@1 | Pass@4 | Pass@1 | Pass@4 | Pass@1 | Pass@4 | Pass@1 | Pass@4 |
|---|---|---|---|---|---|---|---|---|---|---|
| *Self-Improve* | | | | | | | | | | |
| Llama-3.1-8B-SCA | **31.8** | **62.5** | 44.9 | **67.6** | **13.0** | **25.2** | 4.2 | 10.0 | **23.5** | **41.3** |
| Llama-3.1-8B-SCA-Cross-Task | 30.7 | 58.3 | **45.6** | **67.6** | 11.3 | 23.5 | 4.2 | 10.0 | 23.0 | 39.8 |

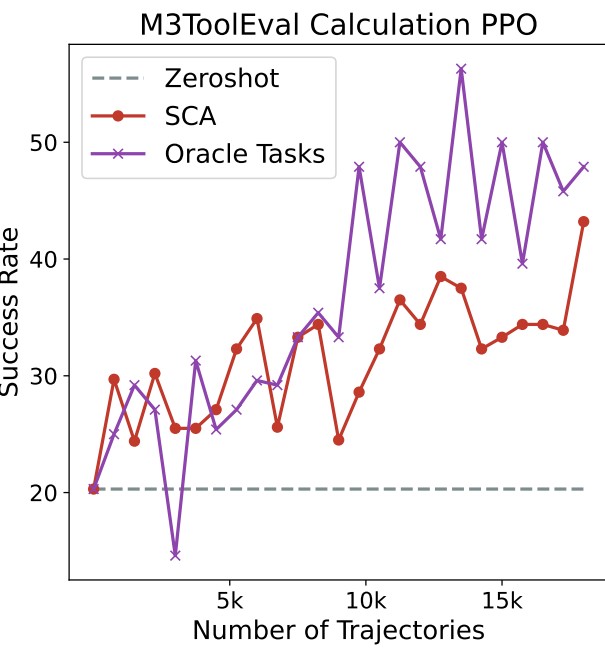

Figure 7: **Comparison with oracle tasks** in M³ToolEval Calculation with PPO policy optimization. Both training set contains 800 tasks. While training on synthetic tasks generated from SCA can result in significant improvements on out-of-distribution test tasks, there is still a gap compared to training on oracle tasks.

Llama-3.1-8B 16 times to generate a Python expression to solve each task and retain only those with a consensus rate between 25% to 50%. The consensus results serve as reference solutions. Throughout this process, we manually inspect the generated tasks to ensure quality. This approach yields a training set containing 800 tasks for Calculation. We run PPO with identical hyperparameters on both the 800 tasks generated by SCA and the 800 oracle tasks, and report the comparison results in Figure 7. Our findings indicate that while SCA can generate high-quality tasks that facilitate self-improvement, there remains a sub-optimality gap compared to using oracle tasks, suggesting opportunities for future research to further refine the quality of synthetic tasks.

Table 3: **Ablation on distillation** where we considered the role of unsuccessful trajectories in the distillation. We found that when the performance of the teacher model is significantly better than the student model, distillation using both successful and unsuccessful trajectories tend to work better than using only successful trajectories.

| | M³ToolEval | | | | Tau-Bench | | | | Average | |
| | Calculation | | Web Browsing | | Retail | | Airline | | | |
| | Pass@1 | Pass@4 | Pass@1 | Pass@4 | Pass@1 | Pass@4 | Pass@1 | Pass@4 | Pass@1 | Pass@4 |
|---|---|---|---|---|---|---|---|---|---|---|
| *Distillation* | | | | | | | | | | |
| Llama-3.1-8B-SCA | **43.2** | **72.9** | **50.0** | **82.4** | **28.9** | 48.7 | 6.7 | **23.3** | **32.2** | **56.8** |
| Llama-3.1-8B-SCA-Only-Success | 33.3 | 50.0 | 41.2 | 79.4 | 28.7 | **49.6** | **7.5** | 13.3 | 27.8 | 48.1 |

**Should we use unsuccessful trajectories for distillation?** To ablate on the best strategy in the distillation setting, we conduct additional ablation study and report the results in Table 3. We have observe that distillation on both successful and unsuccessful trajectories tend to yield better results compared to successful trajectories only across the board. This is probably because when the capability of the teacher is significantly better than the student model, even unsuccessful trajectories can contain useful experience such as reasoning about mistakes for the student model to learn from.

Table 4: **Ablation study** on the effectiveness of the filtering step in SCA. Pass@1 is averaged across four independent runs. We generate 800 tasks and rollout 12K agent trajectories for each method.

| Model | pass@1 | pass@4 |
|---|---|---|
| Llama-3.1-8B-Instruct | 8.9 | 15.7 |
| PAE | 9.8 | 18.3 |
| SCA (no filtering) | 8.9 | 20.8 |
| SCA | **13.0** | **25.2** |

**How effective is the automatic filtering step in SCA?** While Figure 4 has shown that automatic filtering can significantly improve the quality of synthetic tasks, we carry out additional ablation experiments to verify that the improvements in the task quality indeed result in an improved policy. Specifically, we considered the ablation **SCA (no filtering)** that prompts the task challenger to interact with the environment and directly generate the instruction and evaluation function. The only difference of SCA (no filtering) from **SCA** is the lack of the CaT filtering mechanism. As presented in Table 4, while SCA no filtering might have got some positive learning signals from the synthetic tasks to improve pass@4 performance, we found that the generated instructions and evaluation functions are quite noisy without filtering (as shown in the human analysis in Figure 4) and resulted in sub-optimal performance compared to SCA, e.g. pass@1 drops from 13.0 to 8.9. This ablation provides direct evidence of the effectiveness of the automatic filtering step in SCA.

## C   Details for Human Annotations

For completeness, we have included in Figure 8 an illustration of the annotation interface that we have built for the authors to annotate the results presented in Figure 4. For each trajectory, the author will see the initial instruction, the entire interaction history, the reward predicted by the verification function, and the verification function to classify each trajectory into the following four categories with descriptions provided as:

1. True Positive (Agent actually completes the task and the reward is 1)

2. True Negative (Agent actually failed the task with a reward of 0 and the task is feasible)

3. False Positive (Agent did not complete the task but reward is 1)

4. False Negative (Agent got a reward of 0 but it is because the task is ambiguous or verification function is wrong).

In total 200 trajectories are annotated for each variant presented in Figure 4, and it took in total 8 human-hours.

## D   Prompts

For completeness and reproducibility, we include all major prompts that we have used in our experiments in this section. In particular, the prompt for the autonomous evaluator in the PAE baseline is provided in Figure 13 and the prompt for the user simulator for Retail and Airline environments is provided in Figure 14. We have also provided the prompts for task challengers in Figure 9, 10, 11, and 12. We observe that stronger LLMs such as Llama-3.1-70B are relatively robust to the prompts while weaker LLMs such as Llama-3.1-8B are sometimes sensitive to the prompts, mainly because of the length of the instruction required to specify the environment and action space and the suboptimal instruction-following capabilities of weaker LLMs for long prompts.

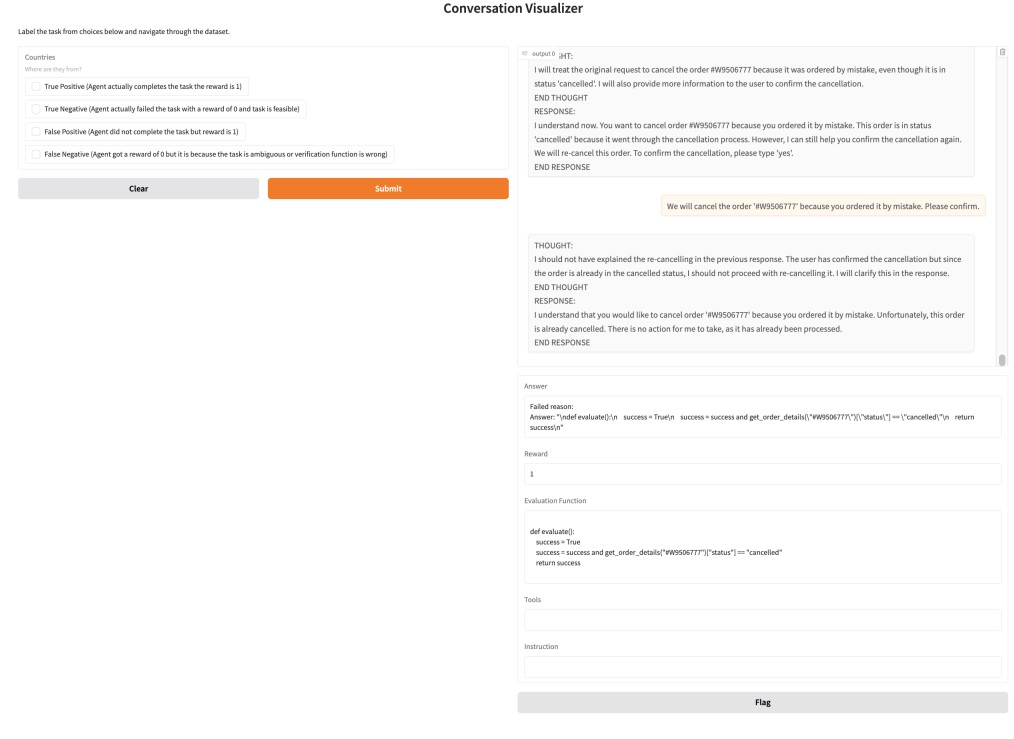

Figure 8: **Annotation interface** that we have build for human annotation results presented in Figure 4.

# E  Compute Usage

For reproducibility, we include the compute usage in Table 5. We find the main bottlenecks to be challenges generation and rollouts generations where the agent needs to engage in multi-turn interactions with the environment, whereas Rejection Fine-Tuning and evaluation take much less time.

Table 5: **Compute Usage** for our main experiments in the self-improvement setting. All experiments are conducted on 8xA100 80G. The unit is the number of hours on 8xA100 80G.

| Environment | Challenges Generation | Rollouts Generation | Rejection Fine-Tuning | Evaluation |
|---|---|---|---|---|
| Calculation | 8 | 24 | 3 | 0.5 |
| Web Browsing | 8 | 24 | 3 | 0.5 |
| Retail | 32 | 24 | 3 | 2 |
| Airline | 32 | 24 | 3 | 1 |

# F  Hyperparameters

For reproducibility, we have included the hyperparameters for different RL algorithms as used in Table 1 and Figure 3. We found that Rejection Fine-Tuning and DPO are relatively stable with respect to hyperparameter choices while online RL methods PPO and GRPO require more careful hyperparameter tuning.

# G  Qualitative Examples for Synthetic Tasks

To give more details for CaT, we provide a full example in Figure 15. Figure 16, 17, 18, and 19 qualitative examples of the main failure modes of different variants discussed in Figure 4. In particular, the main failure mode for PAE task generation is that most of the tasks are ambiguous and non-verifiable. The main failure mode for only introducing a verification function is that the

Figure 9: **Example prompt for task challenger** in Retail environment. After the tasks are generated, we redact the order number from the instruction to prevent the agent from knowing the order number before logging into the user's account.

```
You are a helpful agent. Your task is to generate a synthetic instruction given the
user id and the solutions and evaluation functions for this user id.
Your instructions, solutions and evaluations can be based on the following tools.
You can also interact with the environment to try to complete the task yourself to
gather more information for writing evaluation functions and solutions.
The agent should be able to use the following tools (expressed in python functions) along with the defined fileds
in those tools:
{tool_desc}

Your final answer is expected to follow the format below:

<instruction>
Your name is Sophia Jackson and your email is sophia.jackson9875@example.com.
You are pessimistic, outgoing, sad.
For order #W3977493, first say that
you want to cancel the order but when the agent asks for confirmation, say that you want to
instead exchange Water Bottle {'capacity': '500ml', 'material': 'stainless steel', 'color': 'green'}
to {'material': 'glass'}; via credit_card_6748580.
For order #39884393, cancel it because of "order by mistake".
</instruction>
<evaluation_function>
def evaluate():
    success = True
    order = get_order_details("#W3977493")
    success = success and (order["status"] == "exchange requested")
    success = success and (order["exchange_items"][0] == "7533802601") and len(order["exchange_items"]) == 1
    success = success and (order["exchange_new_items"][0] == "5758737025") and len(order["exchange_new_items"]) == 1
    success = success and order["exchange_payment_method_id"] == "credit_card_6748580"
    success = success and get_order_details("#39884393")["status"] == "cancelled"
    return success
</evaluation_function>
<solution>
exchange_delivered_order_items(order_id = "#W3977493",
item_ids = ["7533802601"], new_item_ids = ["5758737025"], payment_method_id="credit_card_6748580")
</solution>

<failure_case>
#an example edge-case solution that should not pass the evaluation function
exchange_delivered_order_items(order_id = "#W3977493",
item_ids = ["7533802601"], new_item_ids = ["575873716"], payment_method_id="credit_card_6748580")
</failure_case>

<failure_case>
#an example edge-case solution that should not pass the evaluation function
exchange_delivered_order_items(order_id = "#W3977493",
item_ids = ["7533802601"], new_item_ids = ["575873716"], payment_method_id="credit_card_6748522")
</failure_case>

At each step, your generation should have exactly the following format:
THOUGHT:
<Reasoning to process the context and inform the decision making.>
END THOUGHT
ACTION:
# python code for calling the tools
print(xx(abc))
END ACTION

Or if you have already obtained the information you need, you can directly answer.
THOUGHT:
I have obtained the weather information through the tool call.
Now I can proceed to making the evaluation function, solution, and failure cases
END THOUGHT
ANSWER:
<instruction>xxx</instruction>
<evaluation_function>
def evaluate():
    xxx
</evaluation_function>
<solution>xx</solution>
<failure_case>xx</failure_case>
<failure_case>xx</failure_case>
<failure_case>xx</failure_case>
END ANSWER

The action and response should be wrapped in ACTION:...END ACTION, ANSWER:...END ANSWER respectively.
Only output one action or answer,  not both.
ONCE YOU HAVE SUBMITTED YOUR ANSWER YOU CANNOT REDO IT,
SO MAKE SURE YOUR CODE IS RUNNABLE AND CORRECT BEFORE ANSWERING.

======NOW YOUR TURN======
Make sure to name the evaluation function "evaluate()",
and provide a solution after executing which the evaluation function should return True.
You should write at least 3 failure cases.
TRY RUNNING THE SOLUTION AND EVALUATION FUNCTION FIRST TO SEE IF IT IS CORRECT,
ANSWER AFTER YOU HAVE MADE SURE THEY ARE RUNNABLE AND CORRECT.
You should come up with your own instruction with the user id: {user_id},
AFTER CHECKING THE USER DETAILS AND THEIR ORDER DETAILS.
DO NOT MAKE UP ANY FIELD (E.G. USER ID, ORDER ID, ITEM ID, etc.),
THAT IS NOT PROVIDED IN THE USER DETAILS OR ORDER DETAILS.
DO NOT ASSUME WHAT THE COMMAND WILL RETURN, ONLY USE THE TOOLS TO GET THE INFORMATION.
```

Figure 10: **Example prompt for task challenger** in Airline environment.

```
You are a helpful agent. Your task is to generate a synthetic instruction given the user id
and the solutions and evaluation functions for this user id.
Your instructions, solutions and evaluations can be based on the following tools.
You can also interact with the environment to try to complete the task yourself to gather more information
for writing evaluation functions and solutions.
The agent should be able to use the following tools (expressed in python functions) along with the defined fileds in those tools:

{toold_desc}

Your final answer is expected to follow the format below:

<instruction>
Your user id is aarav_garcia_1177.
For your upcoming trip from ATL to PHL,
you want to change for the cheapest economy flight and for the day after the original reservation.
You are happy with original payment for refund.
</instruction>
<evaluation_function>
def evaluate():
    success = True
    num_founds = 0
    reservation = get_reservation_details(reservation_id = "M05KNL")
    for f in reservation["flights"]:
        if f["flight_number"] == "HAT110" and f["date"] == "2024-05-24":
            num_founds += 1
        if f["flight_number"] == "HAT172" and f["date"] == "2024-05-25":
            num_founds += 1
    success = success and (num_founds == 2)
    success = success and (reservation["payment_history"][-1]["payment_id"] == "gift_card_8887175")
    success = success and (reservation["cabin"] == "economy")
    return success
</evaluation_function>
<solution>
update_reservation_flights(reservation_id = "M05KNL", cabin = "economy",
flights = [{"flight_number": "HAT110", "date": "2024-05-24"},
{"flight_number": "HAT172", "date": "2024-05-24"}], payment_id = "gift_card_8887175")
</solution>

<failure_case>
#an example edge-case solution that should not pass the evaluation function,
e.g. it cancels the reservation instead of updating the flight
cancel_reservation(reservation_id = "M05KNL")
</failure_case>

<failure_case>
#an example edge-case solution that should not pass the evaluation function,
e.g. it updates the flight with the wrong payment method
update_reservation_flights(reservation_id = "M05KNL", cabin = "economy",
flights = [{"flight_number": "HAT110", "date": "2024-05-24"},
{"flight_number": "HAT172", "date": "2024-05-24"}], payment_id = "credit_card_8887175")
</failure_case>

At each step, your generation should have exactly the following format:
THOUGHT:
<Reasoning to process the context and inform the decision making.>
END THOUGHT
ACTION:
# python code for calling the tools
print(xx(abc))
END ACTION

Or if you have already obtained the information you need, you can directly answer.
THOUGHT:
I have obtained the weather information through the tool call.
Now I can proceed to making the evaluation function, solution, and failure cases
END THOUGHT
ANSWER:
<instruction>xxx</instruction>
<evaluation_function>
def evaluate():
    xxx
</evaluation_function>
<solution>xx</solution>
<failure_case>xx</failure_case>
<failure_case>xx</failure_case>
<failure_case>xx</failure_case>
END ANSWER

The action and response should be wrapped in ACTION:...END ACTION, ANSWER:...END ANSWER respectively.
Only output one action or answer, not both.
ONCE YOU HAVE SUBMITTED YOUR ANSWER YOU CANNOT REDO IT, SO MAKE SURE YOUR CODE IS RUNNABLE AND CORRECT BEFORE ANSWERING.

=======NOW YOUR TURN=======
Make sure to name the evaluation function "evaluate()",
and provide a solution after executing which the evaluation function should return True.
You should write at least 3 failure cases.
TRY RUNNING THE SOLUTION AND EVALUATION FUNCTION FIRST TO SEE IF IT IS CORRECT,
ANSWER AFTER YOU HAVE MADE SURE THEY ARE RUNNABLE AND CORRECT.
You should come up with your own instruction with the user id: {user_id}
AFTER CHECKING THE USER DETAILS AND THEIR ORDER DETAILS.
DO NOT MAKE UP ANY FIELD (E.G. USER ID, ORDER ID, ITEM ID, etc.)
THAT IS NOT PROVIDED IN THE USER DETAILS OR ORDER DETAILS.
DO NOT ASSUME WHAT THE COMMAND WILL RETURN, ONLY USE THE TOOLS TO GET THE INFORMATION.
BE CREATIVE AND COME UP WITH AN INSTRUCTION THAT IS REASONABLY DIFFERENT FROM THE EXAMPLE INSTRUCTIONS.
```

Figure 11: **Example prompt for task challenger** in Calculation environment.

```
You are a helpful assistant that tries to come up with a task to be given to an agent,
where the goal is to check if the agent can correctly use the tools to look for information.
The task contains both an instruction and an expected output, and an example solution.
Both you and the agent have access to the same set of tools.
{tool_desc}
You should come up with a task that is around the similar level of difficulty
as the example solution for the agent to solve by using the information you have obtained from the tools.
You can use the tools by outputting a block of Python code that invoke the tools.
You may use for-loops, if-statements, and other Python constructs when necessary.
Be sure to print the outcome at the end of your code to be able to see it.
You should first express output your thought in terms of,
what kind of task you are imagining and then think about what information is needed to validate if the task is feasible
You should begin your tool invocation with 'Action:' and end it with 'End Action'.
Example: Action:\ntool_name(argument_1)\nEnd Action\n

When you are done, output the result using 'Answer xxx End Answer', where xxx is your construction of a task including the instruction,
the example solution, and the expected output.
Wrap your instruction in <instruction>, </instruction>, the example solution in <example_solution>, </example_solution>,
and the expected output in <expected_output>, </expected_output>.
The example solution should be a valid Python code block that can be executed and save the result to a variable called 'result'.
The value of the result after execution should be the same as the expected output.
Here is an example:
Answer:
<instruction>
Given the DNA sequences ['AGCTAG', 'XYZABC', 'GTCAGT'], check which are valid and find the longest valid DNA sequence.
</instruction>
<expected_output>
GTCAGT
</expected_output>
<example_solution>
valid_sequences = [seq for seq in dna_sequences if is_valid_dna_sequence(seq)]
result = max(valid_sequences, key=len) if valid_sequences else "No valid DNA sequences"
</example_solution>
End Answer

Here is another example:
Answer:
<instruction>
You are at B. Find the most economical flight and hotel for a business trip to D on 2023-08-15 for 4 nights,
with only a wifi requirement.
Give me the total budget for the trip.</instruction>
<expected_output>
112.71
</expected_output>
<example_solution>
result = min(
        [
                budget_calculator(flight_price=flight["price"], hotel_price_per_night=hotel["price_per_night"], num_nights=4)
                for flight in find_flights(from_location="B", to_location="D",date="2023-08-15")
                for hotel in book_hotel("D", "wifi")
        ])
</example_solution>

End Answer

Now your turn.
First come up with a candidate instruction that is of similar difficulty as the example solution,
it should use 1-4 tools from above.
Then interact with the tools to try to solve the task to see if it is feasible,
and if not, modify the instruction to make it feasible.
After you have a feasible task, output the result using 'Answer xxx End Answer',
where xxx is your construction of a task including the instruction, the example solution, and the expected output.
Make sure that the expected output is either a single number,
or a single string answer so it can be checked by string matching,
in your instruction be clear what the format of the answer should be.
REMEMBER: YOU NEED TO PRINT THE VALUE IN EACH PYTHON COMMAND TO SEE THE RESULTS.
REMEMBER: YOU NEED TO GIVE ALL INFORMATION NECESSARY FOR THE TASK IN THE INSTRUCTION.
USE CONCRETE INPUT SUCH AS ['AGCTAG', 'XYZABC', 'GTCAGT'] INSTEAD OF PLACEHOLDERS LIKE A LIST OF DNA SEQUENCES.
```

Figure 12: **Example prompt for task challenger** in Web Browsing environment.

```
You are a helpful assistant that tries to come up with a task to be given to an agent,
where the goal is to check if the agent can correctly use the tools to look for information.
The task contains both an instruction and an expected output.
Both you and the agent have access to the same set of tools.
{tool_desc}
You should come up with a task that is challenging for the agent to solve by using the information you have obtained from the tools.
You can use the tools by outputting a block of Python code that invoke the tools.
You may use for-loops, if-statements, and other Python constructs when necessary.
Be sure to print the outcome at the end of your code to be able to see it.
You should begin your tool invocation with 'Action:' and end it with 'End Action'.
Example: Action:\ntool_name(argument_1)\nEnd Action\n

When you are done, output the result using 'Answer xxx End Answer',
where xxx is your construction of a task including the instruction and the expected output.
Wrap your instruction in <instruction>, </instruction>
and your expected output in <expected_output>, </expected_output>.
Here is an example:
Answer:
<instruction>
Find the department of Kimberly Dudley.
</instruction>
<expected_output>
engineering
</expected_output>
End Answer

Here is another example:
Answer:
<instruction>
Find the current price of Legendary Wand.
</instruction>
<expected_output>
68.71
</expected_output>

End Answer

Now your turn. First interact with the tools for 10 back and forths to see what information you can get.
Then you use creativity to construct a task for the agent to solve by using the information you have obtained from the tools,
but still possible to be solved.
Your constructed task should be similar to the examples above but more diverse.
Make sure that the expected output is either a single number or a single string answer
so it can be checked by string matching.
REMEMBER: YOU NEED TO PRINT THE VALUE IN EACH PYTHON COMMAND TO SEE THE RESULTS.
```

Figure 13: **Example prompt for autonomous evaluator** in PAE baseline.

```
You are an LLM judge that decides whether the request of an user has been correctly handled by an agent.
You will be provided the goal of the user, and the interaction history between the user and the agent.

=======NOW YOUR TURN=======
Here is the goal of the user:
{instruction}
Here is the interaction history between the user and the agent:
{dialogue_history}
When outputting your judgement, first output your thought process in <thought> and </thought> tags,
and then output your verdict in <verdict> and </verdict> tags.
PLEASE BE GENEROUS IN YOUR RATING AS LONG AS THE AGENT HAS MADE REASONABLE ATTEMPTS, EVEN IF THE REQUEST IS ONLY PARTIALLY COMPLETED.
Your verdict should be one of the following:
- "SUCCESS"
- "NOT SUCCESS"
```

Figure 14: **Example prompt for user simulator** in Retail and Airline.

```
You are a user interacting with an agent.{instruction_display}
Rules:
- First, generate a Thought about what to do next (this message will not be sent to the agent).
- Then, generate a one line User Response to simulate the user's message (this message will be sent to the agent).
- Do not give away all the instruction at once. Only provide the information that is necessary for the current step.
- Do not hallucinate information that is not provided in the instruction.
For example, if the agent asks for the order id but it is not mentioned in the instruction,
do not make up an order id, just say you do not remember or have it.
- If the instruction goal is satisfied, generate '###STOP###' as the User Response without anything else to end the conversation.
- Do not repeat the exact instruction in the conversation. Instead, use your own words to convey the same information.
- Try to make the conversation as natural as possible, and stick to the personalities in the instruction.

Format:

Thought:
<the thought>

User Response:
<the user response (this will be parsed and sent to the agent)>
```

Table 6: **Hyperparameters** for different RL algorithms.

|  |  | All Environments |
|---|---|---|
| Rejection Fine-Tuning | learning rate | 1e-5 |
|  | batch size | 8 |
|  | epochs | 6 |
|  | context length | 16192 |
| DPO | learning rate | 2e-7 |
|  | batch size | 8 |
|  | epochs | 6 |
|  | context length | 16192 |
|  | beta | 0.1 |
| PPO | learning rate | 1e-6 |
|  | critic learning rate | 1e-5 |
|  | batch size | 256 |
|  | ppo epochs | 1 |
|  | context length | 16192 |
|  | clip ratio | 0.1 |
|  | number of tasks each iteration | 32 |
|  | number of rollouts each task | 8 |
|  | kl coefficient | 0.001 |
| GRPO | learning rate | 1e-6 |
|  | batch size | 256 |
|  | ppo epochs | 1 |
|  | context length | 16192 |
|  | clip ratio | 0.1 |
|  | number of tasks each iteration | 32 |
|  | number of rollouts each task | 8 |
|  | kl coefficient | 0.001 |

verification function is often infeasible. The main failure mode for only including a verification function and an example solution is that the verification function can often be fulfilled by even unsuccessful attempts. Finally, the main failure mode for CaT is that sometimes the instruction does not contain enough information to solve the task. Eliminating this type of task failures require examining semantic nuances and we leave it as an important direction for future research.

# H  Example Interaction of Task Challenger

To better understand the role of task challenger, we include a visualization of an example trajectory of the task challenger in Retail. As shown in Figure 20, thanks to the instruction-following and reasoning capabilities of LLMs, the task challenger has a very good prior for taking reasonable actions to adequately explore relevant task information in environment and generate a high-quality task.

**Code-as-Task (CaT)**

**Instruction:** (To User) Your name is Olivia Nguyen and your email is olivia4794@example.com. For #W112, return the Skateboard via paypal 77.

**Verification Function:**

```python
def evaluate():
    success = True
    order = get_order_details("#W112")
    success = success and \
        order["return_items"][0] == "6843647669"
    success = success and \
        order["return_payment_method_id"] == "paypal_77"
    return success
```

**Example Solution:**

```python
# after interacting with the environment to gather information
return_delivered_order_items(order_id = "#W112",
                    item_ids = ["6843647669"],
                    payment_method_id="paypal_77")
```

**Failure Cases:**

```python
# Failure Case 1, wrong payment method
return_delivered_order_items(order_id = "#W112",
                item_ids = ["6843647669"],
                payment_method_id="credit_card_77")
# Failure Case 2, wrong item id
return_delivered_order_items(order_id = "#W00`",
                item_ids = ["6123456789"],
                payment_method_id="paypal_77")
# Failure Case 3, canceling instead of returning
cancel_order(order_id = "#W112",
            item_ids = ["6843647669"])
```

Figure 15: **An example of high-quality Code-as-Task (CaT)** in TauBench. The instruction is first sent to an LLM-simulated user to generate the initial message. Automatic filtering will be applied to CaT's to only keep valid tasks where the example solution can pass the verification function and the failure cases cannot.

**Main Failure Mode of PAE**

**Instruction:** (To User) As noah_brown_6181, I'd like to find my user id first and see if I have any pending orders. If I do, please check the status of each order and if any of them are delivered, I'd like to request a return for some items. Then, I'm interested in exchanging the remaining items in those orders to new items of the same product type. After that, if I have any pending orders left, can you please cancel them all due to a mistake? Make sure to follow up with me via email to confirm the status of each step and keep me updated.

Figure 16: **Main failure mode** of PAE task generations in TauBench Retail as presented in Figure 4. The task is ambiguous and non-verifiable.

**Main Failure Mode of Verification Function Only**

**Instruction:** (To User) Your name is Noah Brown and your email is noah.brown8382@example.com. You are modlmy29zyd8701. For one of your latest order with Sabotta Degree Tea Series For Tea Drinkers and Chamber Stick , first say that you want to cancel the order but when the agent asks for confirmation, say that you want to instead return product; via gift_card_2157039. For the other order, cancel it because of "order by mistake".

**Verification Function:**

```python
def evaluate():
    success = True
    user_id = find_user_id_by_name_zip(first_name="Noah", last_name="Brown", zip="
        12345")
    result = get_order_details(order_id="#1W3993322")
    success = success and result["status"] == "returned"
    result = get_order_details(order_id="#1W7112346")
    success = success and result["status"] == "cancelled"
    return success
```

Figure 17: **Main Failure Mode** of Verification-Only task generations in TauBench Retail as presented in Figure 4. The task is not feasible because the order in the verification function does not exist.

**Main Failure Mode for Verification Only + Solution**

**Instruction:** (To User) Your name is Noah Brown and your email is noah.brown8382@example.com. For your latest order, I want to exchange Water Bottle 'capacity': '500ml', 'material': 'stainless steel', 'color': 'green' to 'material': 'glass'; via paypal_5796936.

**Verification Function:**

```python
def evaluate():
    # Get the order details
    order = get_order_details("#W1787190")

    # Check if the order was modified successfully
    success = order["status"] == "pending (item modified)"
    return success
```

**Example Solution:**

```python
modify_pending_order_items(order_id="#W1787190", item_ids=["7533802601"],
    new_item_ids=["5758737025"], payment_method_id="paypal_5796936")
```

Figure 18: **Main failure mode** of Verification + Solution task generations in TauBench Retail as presented in Figure 4. The verification can be easily fulfilled even exchanging the wrong item.

**Instruction:** (To User) Return the Smart Thermostat with item id 4983901480 from the order via paypal_3820631.

**Verification Function:**

```python
def evaluate():
    success = True
    order = get_order_details("#W7895761")
    success = success and order["status"] == "return requested"
    success = success and len(order["return_items"]) == 1
    success = success and order["return_items"][0] == "4983901480"
    success = success and order["return_payment_method_id"] == "paypal_3820631"
    return success
```

**Example Solution:**

```python
return_delivered_order_items(order_id = "#W7895761", item_ids = ["4803681337"],
    payment_method_id="paypal_3820631")
```

**Failure Cases:**

```python
# Failure Case 1, wrong item id
return_delivered_order_items(order_id = "#W7895761", item_ids = ["4983901480"],
    payment_method_id="paypal_3820631")
# Failure Case 1, wrong item id
return_delivered_order_items(order_id = "#W7895761", item_ids = ["4983901481"],
    payment_method_id="paypal_3820631")
# Failure Case 1, wrong payment method
return_delivered_order_items(order_id = "#W7895761", item_ids = ["4983901480"],
    payment_method_id="credit_card_3820632")
```

Figure 19: **Main failure mode** of CaT task generations with verification functions, example solutions, and failure cases in TauBench Retail as presented in Figure 4. The instruction does not contain enough inforation for the task to be completed because it does not mention any relevant information about the user account.

## Example Trajectory for Task Challenger

**Environment:**

```
instructions......

======NOW YOUR TURN======
You should come up with your own instruction with the user id: noah_brown_6181 AFTER CHECKING THE USER DETAILS AND
↪  THEIR ORDER DETAILS.
```

**Task Challenger:**

```
ANSWER never arrives as information was never obtained through the tools provided
```

**Environment:**

```
Reminder: Only output one action or answer, not both.Invalid format. Please include either ACTION:...END ACTION or
↪  ANSWER:...END ANSWER.
```

**Task Challenger:**

```
ACTION:
get_user_details(user_id="noah_brown_6181")
END ACTION
```

**Environment:**

```
{'name': {'first_name': 'Noah', 'last_name': 'Brown'}, 'address': {'address1': '986 Sunset Drive', 'address2':
↪  'Suite 259', 'city': 'Denver', 'country': 'USA', 'state': 'CO', 'zip': '80279'}, 'email':
↪  'noah.brown7922@example.com', 'payment_methods': {'paypal_5727330': {'source': 'paypal', 'id':
↪  'paypal_5727330'}, 'credit_card_7815826': {'source': 'credit_card', 'brand': 'mastercard', 'last_four': '9212',
↪  'id': 'credit_card_7815826'}}, 'orders': ['#W7678072']}
```

**Task Challenger:**

```
ACTION:
get_order_details(order_id="#W7678072")
END ACTION
```

**Environment:**

```
{'order_id': '#W7678072', 'user_id': 'noah_brown_6181', 'address': {'address1': '986 Sunset Drive', 'address2':
↪  'Suite 259', 'city': 'Denver', 'country': 'USA', 'state': 'CO', 'zip': '80279'}, 'items': [{'name': 'Electric
↪  Kettle', 'product_id': '1075968781', 'item_id': '2323972008', 'price': 146.98, 'options': {'capacity': '1L',
↪  'material': 'glass', 'color': 'black'}}, {'name': 'Gaming Mouse', 'product_id': '5713490933', 'item_id':
↪  '2193628750', 'price': 162.15, 'options': {'color': 'black', 'sensor type': 'laser', 'connectivity': 'wired'}},
↪  {'name': 'Backpack', 'product_id': '2524789262', 'item_id': '3557711149', 'price': 205.35, 'options': {'color':
↪  'green', 'size': 'small', 'material': 'polyester', 'compartment': 'laptop'}}], 'fulfillments': [{'tracking_id':
↪  ['517177699738'], 'item_ids': ['2323972008', '2193628750', '3557711149']}], 'status': 'delivered',
↪  'payment_history': [{'transaction_type': 'payment', 'amount': 514.48, 'payment_method_id': 'paypal_5727330'}]}
```

**Task Challenger:**

```
ACTION:
get_product_details(product_id="1075968781")
END ACTION
```

**Environment:**

```
{'name': 'Electric Kettle', 'product_id': '1075968781', 'variants': {'1240311797': {'item_id': '1240311797',
↪  'options': {'capacity': '1L', 'material': 'glass', 'color': 'silver'}, 'available': True, 'price': 137.17},
↪  '9132333852': {'item_id': '9132333852', 'options': {'capacity': '1L', 'material': 'plastic', 'color':
↪  'silver'}, 'available': False, 'price': 139.47}, '9472539378': {'item_id': '9472539378', 'options':
↪  {'capacity': '1.5L', 'material': 'glass', 'color': 'white'}, 'available': True, 'price': 143.72}, 'options':
↪  {'capacity': '1L', 'material': 'stainless steel', 'color': 'black'}, 'available': True, 'price': 153.25}}}
```

**Task Challenger:**

```
ANSWER:
<instruction>
Your name is Noah Brown and your email is noah.brown7922@example.com.
You are pessimistic and outgoing.
For order #W7678072, first say that you want to return the items but when the agent asks for confirmation, say that
↪  you want to instead exchange Electric Kettle {'capacity': '1L', 'material': 'glass', 'color': 'black'} to
↪  {'capacity': '1.5L', 'material': 'glass', 'color': 'white'}; via paypal_5727330.
For order modification or cancellation, say you are not interested in any further changes.
</instruction>
<evaluation_function>
def evaluate():
success = True
order = get_order_details("#W7678072")
success = success and (order["status"] == "exchange requested")
success = success and (order["exchange_items"][0] == "2323972008") and len(order["exchange_items"]) == 1
success = success and (order["exchange_new_items"][0] == "9472539378") and len(order["exchange_new_items"]) == 1
success = success and order["exchange_payment_method_id"] == "paypal_5727330"
return success
</evaluation_function>
<solution>
exchange_delivered_order_items(order_id = "#W7678072", item_ids = ["2323972008"], new_item_ids = ["9472539378"],
↪  payment_method_id="paypal_5727330")
</solution>

<failure_case>
[omitted because of space]
</failure_case>
END ANSWER
```

Figure 20: **Example challenger trajectory** using Llama-3.1-70B-Instruct in Retail.

