# OpenReview forum: "Self-Challenging Language Model Agents"
_NeurIPS.cc/2025/Conference — NeurIPS 2025 poster_

### Official Review · Reviewer_ndna · 2025-06-04

**Clarity:** 2
**Significance:** 3
**Originality:** 3
**Rating:** 4
**Confidence:** 4

**Summary:**

This paper proposes Self-Challenging Agents (SCA), a framework for training multi-turn, tool-using language agents using self-generated tasks. The agent plays both a challenger role, where it generates novel tasks through environment interaction, and an executor role, where it learns from those tasks via reinforcement learning. To ensure task quality, the paper introduces a Code-as-Task (CaT) format, which combines an instruction, a verification function, an example solution, and failure cases. The method is evaluated on M3ToolEval and TauBench benchmarks in both self-improvement and distillation settings, showing promising performance gains without relying on existing tasks.

**Questions:**

Can you include an explicit ablation study comparing model performance with vs. without CaT filtering (e.g., training on unfiltered tasks)?

Please clarify how the same model can generate and solve tasks that are both “challenging” and beneficial, if it already knows the answer.

Consider standardizing terminology and formatting (e.g., TauBench vs. taubench), and clearly explain the numeric labels in Figure 4.

If possible, report reward or success rate distributions of CaT-filtered vs. unfiltered tasks, to provide stronger support for the value of the filtering pipeline.

If the rebuttal includes direct evidence that filtering leads to performance improvements (e.g., via ablations), and clearly explains how the challenger/executor dynamic leads to new capabilities rather than repetition, I would be open to raising the score.

**Ethical Concerns:**

["NO or VERY MINOR ethics concerns only"]

**Final Justification:**

Concerns about the paper including evidence of filtering effectiveness and challenger & resolver dynamics have been addressed. If the final version clearly explains the underlying intuition and includes the experimental results to demonstrate the effectiveness of the proposed  methods, I will raise my score to 4.

**Limitations:**

The paper contains a dedicated Limitations section that acknowledges key challenges such as false negatives and environment-specific overfitting.

**Paper Formatting Concerns:**

No major formatting issues

**Quality:**

3

**Strengths And Weaknesses:**

**Strengths:**

The paper tackles an important problem: scalable task synthesis for multi-turn agent training, without manual annotations.

The proposed CaT format is a creative attempt to ensure verifiability and difficulty of synthesized tasks.

Empirical results suggest notable gains in both distillation and self-improvement settings, particularly in partially observable environments.

**Weaknesses:**

The central claim is that the filtering process (i.e., verifying the CaT components) leads to higher-quality tasks and better agent performance. However, no direct ablation on task filtering vs. not filtering is presented. The closest evidence (Table 4 / Figure 4) involves human-labeled task quality and difficulty distributions, which only indirectly support the claim.

The filtering relies on the challenger model generating a correct solution. But in the self-improvement setting, the challenger and executor are the same model. If the model can already generate valid solutions, it’s unclear why these tasks would lead to further improvement. There might need a clear explanation of how the model benefits from solving tasks it can already solve.

There are some writing inconsistencies, such as the inconsistent use of “TauBench” vs. “taubench”, “Verification Only” vs. “verification-function only”. Some visual elements (e.g., Figure 4's “100% / 47.7% / 9.5%” labels) are not adequately explained in the text.

---

> ### Author Rebuttal · Authors · 2025-07-31
>
> Thank you for your review and feedback on the paper. To address your concerns, we have collected additional experiment results with direct evidence of performance improvements and explain the dynamics between challenger and executor.
>
>
> ## Direct evidence that filtering leads to performance improvements
> To address your concern, we have included an explicit study comparing the model performance with and without CaT filtering. Specifically, we considered the ablation “SCA no filtering” that prompts the task challenger to interact with the environment and directly generate the instruction and evaluation function. The only difference of “SCA no filtering” from “SCA” is the lack of the CaT filtering mechanism. We generate 800 tasks and rollout 12K agent trajectories for each method. As shown in the following table, while SCA no filtering might have got some positive learning signals from the synthetic tasks to improve pass@4 performance, we found that the generated instructions and evaluation functions are quite noisy without filtering (as shown in the human analysis in Figure 4) and resulted in sub-optimal performance compared to SCA, e.g. pass@1 drops from 13.0 to 8.9. This shows that the filtering proposed in our paper is important.
>
>
> Tau-Bench Retail
> | Model               | pass@1 (average over four runs) | pass@4 |
> |---------------------|--------|--------|
> | Llama-3.1-8B-Instruct        | 8.9    | 15.7   |
> | PAE                 | 9.8    | 18.3   |
> | SCA no filtering    | 8.9    | 20.8   |
> | SCA                 | 13.0   | 25.2   |
>
> ## Dynamics between challenger and executor
> We respectively disagree that the inclusion of an example solution in CaT limits the potential of self-improvement. In SCA, self-improvement can happen in at least two different ways:
> **Many tasks are easier to verify than proposing a solution** (https://www.jasonwei.net/blog/asymmetry-of-verification-and-verifiers-law) so it might be easier for the agent to come up with an evaluation function than proposing a solution. Once the task challenger has an evaluation function, it can try to “guess” the solution even if it does not know how to solve the problem. Thanks to the filtering mechanism in CaT, the wrong guesses are filtered and the valid tasks that remain can be used to reinforce the correct guesses when the same model serves as the executor. In this way, the executor keeps practicing by being rewarded for correct guesses until eventually it knows how to solve the problem and does not need to guess any more. Hence, the example solutions help it find what it can improve.
> **Hindsight Experience Replay** (https://arxiv.org/abs/1707.01495) has historically been a successful self-improvement method over many applications. The core idea is that the agent might accidentally hit some goals through explorations and we can use the goals that the agent accidentally hits as “hindsight experience” to improve the goal-reaching capability of the agent. SCA can operate in the same way as Hindsight Experience Replay. In the retail environment, the challenger constructing the task might randomly explore some user information in the database and try to run some commands such as exchanging a water bottle to a different size. If any one of the commands succeed, a confirmation message will be returned saying that the water bottle has been exchanged successfully. Based on the confirmation, the challenger knows that some changes have been made and can generate a user instruction in hindsight such as exchange the water bottle for John. But completing the task in the forward way might be much harder where the agent needs to first intentionally figure out the user id for John, and use the user id to find the transaction that contains a water bottle, and checks what sizes are available.
>
>
> ## There are some writing inconsistencies, such as “Taubench” and “taubench”.
> Thanks for spotting those inconsistencies, we will fix them in the next version.

---

> > ### Comment · Reviewer_ndna · 2025-08-03
> >
> > Thank you for your reply. Most of my concerns have been addressed. If the final version clearly explains the underlying intuition and includes the experimental results to demonstrate the effectiveness of your methods, I will raise my score to 4.

---

### Official Review · Reviewer_mkt7 · 2025-06-29

**Clarity:** 3
**Significance:** 3
**Originality:** 3
**Rating:** 5
**Confidence:** 2

**Summary:**

This work tailors a framework to generate high-quality tasks for training LLM-based agents, aiming at alleviating the human efforts required to create such tasks. Agents in this framework perform two distinct roles, i.e., challenger to interacts in an unknown environment with tools to gather information and generate possible tasks, and executor to accomplish these generated tasks by interacting with the same environment. Code-as-Task and several mechanisms are incorporated to ensure the quality of generated tasks.

**Questions:**

The paradigm in which the challenger generates questions for the executor to solve reminds me of Generative Adversarial Networks (GANs), where a major challenge lies in the need for both the generator and the discriminator to improve simultaneously, Otherwise, a large gap in capability between the two can lead to suboptimal training results. In this work, I have only observed improvements to the executor, raising the question of whether this work might encounter similar issues.

**Ethical Concerns:**

["NO or VERY MINOR ethics concerns only"]

**Final Justification:**

My previous score is accurate.

**Limitations:**

yes

**Quality:**

3

**Strengths And Weaknesses:**

1. The proposed method is novel and targets a practically valuable issue in training LLM-based agents.
2. This manuscript is well-organized with clear illustrations, making it easy to follow.

### Weaknesses
1. The experiments were conducted with only one model scale and one model architecture, which raises concerns about the generalizability of the proposed approach. Models of different sizes and architectures possess varying initial capabilities, so it remains unclear whether this method can be applied to a broader range of scenarios.
2. The discussion of failure cases is insufficient. More analysis is needed on the scenarios in which the proposed method fails to perform effectively.
3. More comparisons with existing methods in terms of computational cost are needed to demonstrate the practicality of the proposed approach.

---

> ### Author Rebuttal · Authors · 2025-07-31
>
> Thank you for your review and positive feedback on the paper. We address your concerns below:
>
>
> ## The experiments were conducted with only one model scale and one model architecture, which raises concerns about the generalizability of the proposed approach.
> Because of compute and GPU memory constraints, we were only able to perform experiments at 7B scale. However, in our empirical results, SCA has been shown to improve agent performances across different environments (Table 1), across distillation and self-improvement settings (Table 1), across data scales (Figure 6), and across RL loss choices (Figure 3). We believe that these experiments already provide evidence for the generalizability of SCA.
>
> ## The discussion of failure cases is insufficient.
> Because of space constraints, we have deferred additional analysis of failure cases to the appendix (Appendix B and H, including examples in Figs 16,17, 18 and 19) and also referred to them in a dedicated Limitations Section (Section 6). Figure 4 shows that there is still a non-trivial percentage of False Negative samples and Appendix B demonstrates that there is still a sub-optimality gap between synthetic tasks from SCA and manually crafted tasks. We hope the discussion of these limitations can provide directions for future research. In the camera ready version with additional one page space we can move more of this discussion to the main paper.
>
> ## More comparisons with existing methods in terms of computational cost are needed to demonstrate the practicality of the proposed approach.
> The computational cost of SCA is comparable to other self-improvement methods like PAE because the main computation overhead lies in collecting agent rollouts to perform RL instead of the task proposals. We will provide more details on this in the camera ready version.
>
>
> ## In this work, I have only observed improvements to the executor, raising the question of whether this work might encounter similar issues.
> We also believe that such self-play would be a very promising future direction so that the executor and challenger can improve together. In our experiments, we did observe that the executor started to saturate on the tasks the challengers can offer. We believe this is an exciting research direction for future work.

---

> > ### Comment · Reviewer_mkt7 · 2025-08-06
> >
> > Thanks for your response. My attitude toward this work remains unchanged.

---

### Official Review · Reviewer_BRDP · 2025-06-30

**Clarity:** 3
**Significance:** 3
**Originality:** 3
**Rating:** 4
**Confidence:** 3

**Summary:**

The authors proposed the Self-Challenging framework that trains an agent on high-quality tasks that are generated by itself. There are two roles for the agent: task challenger that gather info and possible tasks,  and task executor that are trained with these tasks. The evaluation on two agent benchmarks "M3ToolEval" and "TauBench", shows the improvements with self-generated training data.

**Questions:**

1. There are four different categories for these generated trajectories: false negative, false positive, true positive, true negative. In the appendix, the false negative is defined as "Agent got a reward of 0 but it is because the task is ambiguous or verification function is wrong". And False positive is "Agent did not complete the task but reward is 1". How about the case that "the reward is 1 and verification function is wrong"?

2. In Table 3, the ablation study shows on successful trajectories only lead worse performance. What is the ration of successful trajectories? the lower performance can be a consequence of un-sufficient samples in the training. It seems hard to conclude that unsuccessful are useful.

3. The plot in Figure 6 is not smooth. Are multiple runs in the experiments?

**Ethical Concerns:**

["Major Concern: Data quality and representativeness"]

**Final Justification:**

The rebuttal solves most of my previous concerned parts (for example, training details). However, I am still concerns about the correctness of the syntactic "verifier". Maybe it is not a big issue for complex tasks, however, it is highly risky parts. Although the results (one Retail environment of one model ) in Figure 4 already show the reduced False Positive and False Negative rates, only two multi-turn tool-use LLM agent benchmarks are tested. It is still lack the strong convincing evidence. More related results would be more convincing.

**Limitations:**

No. Some discussions on how to prevent high risks of generated data can be done.

**Quality:**

3

**Strengths And Weaknesses:**

Strengths:
1. This work proposes a framework that can interact with environment can generate training data by itself. In addition, large improvements are obtained with self-generated training data after RL training. The framework is completed and shows the consist results on two agent benchmark.

2. There are several good interesting analysis in the experiments: comparison between self-improvement and distillation; the effect of different RL algorithms; the impact of different components of framework CaT; scale effect of the number of tasks or the number of trajectories per task etc. All of these analysis enrich the contribution of this work.

Weaknesses:
1. When checking deeper into two different settings (self-improvement and distillation), two different training loss are used. According to Table 1, the distillation setting achieves higher results. In this setting, a stronger LLM are used to sample a dataset among tasks in framework CaT, then, SFT are used in the sampled dataset.  Is it unclear whether SFT loss can be better in Self-improvement loss (RL algorithm).

2. In this proposed framework, as claimed by the authors, a non-trivial percentage of  false negative example can be generated. intuitively , false positive samples can hurt the model seriously. Although as shown in Figure 4, the false positive ratio is very low in some environment (Retail environment). However, it is hard to guarantee whether the low false positive ratio in other environments. What are the possible ways to prevent these risks?

---

> ### Author Rebuttal · Authors · 2025-07-31
>
> Thank you for your review and positive feedback on the paper. We address your concerns below:
>
>
> ## It is unclear whether SFT loss can be better in Self-improvement loss (RL algorithm).
> We believe that there is a misunderstanding here. While indeed we adopted REINFORCE as our primary RL algorithm for the self-improvement setting, because of the reward structure of CaT in our experiments it is actually equivalent to SFT. As explained in Line 166-168, “Because of our reward structure with 0/1 outcome reward, this RL optimization objective is essentially equivalent to performing supervised finetuning (SFT) on successful trajectories (i.e. Rˆ 167 cˆ(sT , aT ) = 1) only, i.e. Rejection Fine-Tuning”. Importantly, however, comparisons of more RL losses – DPO, PPO and GRPO – can be found in Figure 3.
>
> ## What are the possible ways to prevent these risks of false positives?
> An important component of SCA is generating failure cases when creating the task – which we have found to significantly reduce false positives, see Figure 4. When the model generates failure cases, it can try to come up with as many corner test cases and failure modes as possible. CaT validation procedure ensures that the final tasks pass the tests of these corner test cases, so that during agent rollouts those corner cases will not be accidentally classified as success. Intuitively, the more corner test cases there are, the more robust the task should be against these false positives.
>
> ## How about the reward is 1 and verification is wrong?
> In our understanding, for the case of the reward is 1 and verification is wrong seems to mean that the agent got a reward 1 but it should get a reward of 0, meaning that the agent did not finish the task. This case seems to be included in "Agent did not complete the task but reward is 1", studied in Fig 4 and Sec 5.3 (see also Appendix C for annotation details).
>
>
> ## It seems hard to conclude that unsuccessful are useful from experiments in Table 3.
> Indeed, there is a confounding factor in this appendix experiment that there are only about 30% successful trajectories compared to all trajectories, if we keep the number of interactions the same for fair comparison. As a result, we refrained from making general conclusions in the appendix and only report this experiment result as an observation that we have found when the performance of the teacher model is significantly better than the student model. Further study of these aspects in future work would be good!
>
> ## The plot in Figure 6 is not smooth. Are multiple runs in the experiments?
> Because of the compute constraint and the fact that we need to train a separate model for each individual data point, we were unable to perform multiple runs to produce a smoother curve. However, from the 16 data points that we have gathered in Figure 6 with different numbers of training trajectories and training tasks, it already underscores the effectiveness of scaling the number of tasks through SCA compared to solely scaling the number of trajectories with the same tasks.

---

> > ### Comment · Reviewer_BRDP · 2025-08-05
> > **Thanks for the com**
> >
> > "What are the possible ways to prevent these risks of false positives?"
> >  CaT validation procedure ensures that the final tasks pass the tests. The verification method is proposed by challenger (part of CaT system). the However, the verifier is also generated by CaT. How do you verify the correctness of the "verifier"?
> >
> >
> > "How about the reward is 1 and verification is wrong?"
> > Actually, I am thinking about this case: the verification process is passed. However, it is not reliable because the "verifier" is generated synthetically. Are the data results in Figure 4 human annotated or just from the synthetic "verifier" ?

---

> > > ### Author Response · Authors · 2025-08-05
> > >
> > > Thanks for the discussions. We answer your questions below:
> > >
> > > > "What are the possible ways to prevent these risks of false positives?" CaT validation procedure ensures that the final tasks pass the tests. The verification method is proposed by challenger (part of CaT system). the However, the verifier is also generated by CaT. How do you verify the correctness of the "verifier"?
> > >
> > >
> > > While it is not possible to 100% guarantee that the task is completely correct. CaT validation procedure aims to improve the quality of synthetic tasks by verifying from as many perspectives as possible (e.g. example solution and failure cases). As we can see from the human annotation results in Figure 4, CaT can indeed significantly reduce both of False Positive and False Negative rates. This improvement is also reflected in the baseline comparison results in Table 1, where SCA significantly outperforms naive synthetic task generations without CaT filtering mechanisms.
> > >
> > >
> > > >"How about the reward is 1 and verification is wrong?" Actually, I am thinking about this case: the verification process is passed. However, it is not reliable because the "verifier" is generated synthetically. Are the data results in Figure 4 human annotated or just from the synthetic "verifier" ?
> > >
> > > Results in Figure 4 are human annotated. We will make this more clear in the later version

---

### Official Review · Reviewer_Noo7 · 2025-07-03

**Clarity:** 3
**Significance:** 3
**Originality:** 3
**Rating:** 5
**Confidence:** 4

**Summary:**

This paper proposes the framework of Self-Challenging Agents (SCA), enabling agents to generate high-quality tasks by themselves and conduct training. By introducing the "Code-as-Task" (CaT) form, ensure the feasibility, verifiability and difficulty of generating tasks. In multiple rounds of tool usage scenarios, the SCA framework has achieved significant performance improvements in both distillation and self-improvement Settings, demonstrating its potential in reducing human supervision and enhancing the autonomous learning ability of agents.

**Questions:**

Pleade see weaknesses for details.

**Ethical Concerns:**

["NO or VERY MINOR ethics concerns only"]

**Limitations:**

yes

**Quality:**

3

**Strengths And Weaknesses:**

**Strengths**

1. This paper presents a novel SCA framework and CaT form, addressing the issue of difficult task quality assurance in existing task generation methods and providing an effective approach for the self-improvement of LLM agents.

2. Extensive experiments were conducted in multiple multi-round tool usage environments, and significant performance improvements were achieved compared with the baseline method, verifying the effectiveness and practicality of the proposed method.

**Weaknesses**

1. The implementation details of the CaT architecture need further clarification. For instance, does the task challenger go through multiple rounds of feedback processes when generating the validation function? And how does the cat framework handle the situation where the task challenger is unable to provide the correct instance solution?

2. It needs to be compared with other similar self-improvement frameworks, such as Self-Adapting Language Models

---

> ### Author Rebuttal · Authors · 2025-07-31
>
> Thank you for your review and positive feedback on the paper. We address your concerns below:
>
>
> ## The implementation details of the CaT architecture need further clarification.
>
> Yes, if the task challenger fails to construct a valid task, it will receive feedback from the environment. If the task challenger fails to construct a valid task within 10 turns, this trajectory where the proposer interacts with the environment will be filtered out with no valid task. Indeed we noticed in our experiments that sometimes this can be as low as a 5% valid task construction rate for the task challenger as shown in Figure 4, but CaT still leads to significant self-improvement results.
>
> Because of the space constraints, a lot of additional experiment details were deferred to the appendix. For instance, Appendix A includes additional environment-specific implementations of CaT and Appendix I provides an example of the task challenger interaction trajectory. We will move more explanations into the main text in the camera ready version using the extra page of space.
>
> ## It needs to be compared with other self-improvement framework, such as Self-Adapting Language Model
>
> In Table 1, we do have comparisons with the state-of-the-art self-improvement framework for agents PAE at the time of submission, and we have shown the advantage of SCA. While Self-Adapting Language Model is also related to self-improvement, it is released on June 12th – after the submission date – and it has not been tested in the agent domain.

---

> > ### Comment · Reviewer_Noo7 · 2025-08-05
> >
> > Thank you for your response. After considering your replies and the comments from other reviewers, I will maintain my original score.

---

### Decision · Program_Chairs · 2025-09-17

**Decision:**

Accept (poster)

**Comment:**

This paper introduces self-challenging agents, a framework for enabling language model agents to generate their own training tasks. The framework incorporates a code-as-task format, which ensures verifiability and difficulty by combining instructions, verification functions, solutions, and failure cases. The agent plays dual roles as challenger and executor, and is evaluated in both self-improvement and distillation settings on two multi-turn tool-use benchmarks (M3ToolEval and TauBench), achieving consistent performance gains.

**Strengths:**
1. Address an important and practical challenge of scalable task synthesis for multi-turn agents without manual labels;
2. A novel CaT formulation that improves task quality and reduces supervision;
3. Comprehensive experiments and analyses that highlight improvements across different training strategies.

**Weaknesses:**
1. Only one model scale and two benchmarks are tested;
2. Insufficient comparisons with related self-improvement frameworks and computational efficiency;
3. Limited discussion of failure cases and risks (e.g., false positives in task verification); and
4. Certain writing and presentation inconsistencies.

**Conclusion:**

Overall, despite some weaknesses, I agree with the reviewers that the paper makes a meaningful contribution by proposing a novel and practical framework for self-improving agents, with promising empirical results. Therefore, I recommend acceptance. The final revision should include broader experimental validation, stronger ablations, and clearer explanation of the dynamics between challenger and executor roles.